# Radiosensitizing Chemotherapy (Irinotecan) with Stereotactic Body Radiation Therapy for the Treatment of Inoperable Liver and/or Lung Metastases of Colorectal Cancer

**DOI:** 10.3390/cancers13020248

**Published:** 2021-01-11

**Authors:** Loïg Vaugier, Xavier Mirabel, Isabelle Martel-Lafay, Séverine Racadot, Christian Carrie, Véronique Vendrely, Marc-André Mahé, Hélène Senellart, Jean-Luc Raoul, Loïc Campion, Emmanuel Rio

**Affiliations:** 1Department of Radiation Oncology, Institut de Cancérologie de l’Ouest, 44800 St-Herblain, France; loig.vaugier@ico.unicancer.fr (L.V.); ma-mahe@baclesse.unicancer.fr (M.-A.M.); 2Department of Radiation Oncology, Centre Oscar Lambret, 59000 Lille, France; x-mirabel@o-lambret.fr; 3Department of Radiation Oncology, Institut Léon Bérard, 69008 Lyon, France; isabelle.martel-lafay@lyon.unicancer.fr (I.M.-L.); severine.racadot@lyon.unicancer.fr (S.R.); christian.carrie@lyon.unicancer.fr (C.C.); 4Department of Radiation Oncology, Centre Hospitalo-Universitaire Hôpital Saint André, 33000 Bordeaux, France; veronique.vendrely@chu-bordeaux.fr; 5Department of Medical Oncology, Institut de Cancérologie de l’Ouest, 44800 St-Herblain, France; helene.senellart@ico.unicancer.fr (H.S.); jean-luc.raoul@ico.unicancer.fr (J.-L.R.); 6Department of Biostatistics, Institut de Cancérologie de l’Ouest, 44800 St-Herblain, France; loic.campion@ico.unicancer.fr; 7Centre de Recherche en Cancérologie Nantes-Angers (CRCNA), UMR 1232 Inserm—6299 CNRS, Institut de Recherche en Santé de l’Université de Nantes, 44000 Nantes, France

**Keywords:** colorectal cancer, stereotactic radiotherapy, oligometastases, irinotecan, liver metastases, lung metastases

## Abstract

**Simple Summary:**

Stereotactic body radiotherapy (SBRT) is a recognized treatment for liver or lung metastases, but radiosensitivity of colorectal cancer could be lower than other primary cancers. We postulated that local responses could be improved by SBRT with a concomitant radiosensitizing agent (irinotecan). RADIOSTEREO-CAMPTO was a prospective multi-center phase 2 trial conducted between 2008 and 2013. We confirmed that SBRT-Irinotecan was a short, effective and well-tolerated treatment, with no worsening of the quality of life. It allowed for several months of chemotherapy-free periods despite most patients receiving multiple prior lines of treatment. Radiosensitizing irinotecan was able to compensate for lower SBRT dose than nowadays used for liver and lung metastases and could be an interesting regimen in case of tumour-surrounding healthy tissues requiring limited radiation dose.

**Abstract:**

*Background*: Stereotactic body radiotherapy (SBRT) is a recognized treatment for colorectal cancer (CRC) metastases. We postulated that local responses could be improved by SBRT with a concomitant radiosensitizing agent (irinotecan). *Methods*: RADIOSTEREO-CAMPTO was a prospective multi-center phase 2 trial investigating SBRT (40–48 Gy in 4 fractions) for liver and/or lung inoperable CRC oligometastases (≤3), combined with two weekly intravenous infusions of 40 mg/m^2^ Irinotecan. Primary outcome was the objective local response rate as per RECIST. Secondary outcomes were early and late toxicities, EORTC QLQ-C30 quality of life, local control and overall survival. *Results*: Forty-four patients with 51 lesions (liver = 39, lungs = 12) were included. Median age was 69 years (46–84); 37 patients (84%) had received at least two prior chemotherapy treatments. Median follow-up was 48.9 months. One patient with two lung lesions was lost during follow-up. Assuming maximum bias hypothesis, the objective local response rate in ITT was 86.3% (44/51—95% CI: [76.8–95.7]) or 82.4% (42/51—95% CI: [71.9–92.8]). The observed local response rate was 85.7% (42/49—95% CI: [75.9–95.5]). The 1 and 2-year local (distant) progression-free survivals were 84.2% (38.4%) and 67.4% (21.3%), respectively. The 1 and 2-year overall survivals were 97.5% and 75.5%. There were no severe acute or late reactions. The EORTC questionnaire scores did not significantly worsen during or after treatment. *Conclusions*: SBRT with irinotecan was well tolerated with promising results despite heavily pretreated patients.

## 1. Introduction

Treatment of metastatic colorectal cancer (CRC) is the archetype for the progress in multidisciplinary combined systemic and local therapies [1,2]. In the case of oligometastatic disease [1], the uncertainty between localized and microscopic diffuse disease is at the center of the debate regarding the optimal therapeutic treatment sequence [3,4]. Oligometastases present a tremendous challenge for local metastases-directed treatments. Although systemic treatments remain central to standard care, an effective and durable local control of such cases could offer patients a larger chemotherapy-free interval [5,6,7,8] and some cases a cure [3,5]. The combination of local and systemic treatment is currently the standard of care since both the macroscopic and microscopic disease could be targeted at the same time [9], and exclusive metastases-directed treatments can also be proposed for well-selected patients [4].

Surgical resection remains the gold standard for local metastases-directed treatments but is limited to cases of resectable lesions of operable patients [1], whereas the role of perioperative chemotherapy is minimal and still debated [9]. Over the last two decades the landscape for other effective local treatments has grown with the emergence of a large multimodal “toolbox” [1] including radiofrequency ablation (RFA), cryotherapy, microwave ablation, electroporation or stereotactic body radiotherapy (SBRT) [2,10]. With fewer limitations than RFA regarding the tumor size and the vascular vicinity, SBRT is frequently employed for CRC metastases despite the lack of randomized trials [7,10,11,12,13]. Several studies recently reported the efficacy and safety of SBRT for liver and lung metastases of various primary cancers, with an average 1- and 2-year local control around 70–90% and 60–70%, respectively [7,12,13]. The acute and late toxicities of this treatment were globally low. No worsening of the quality of life was noted [14]. It is important to note that most of these studies considered metastases of various primary cancers with heterogeneous radiosensitivies, whereas radiosensitivity of CRC metastases could be lower than other primary cancers [13,15].

Several studies have investigated the benefit of chemotherapy as a radiosensitizing agent in order to increase the local control, e.g., after liver radiotherapy [16,17,18]. The three major drugs for metastatic CRC (5-fluorouracil, oxaliplatin and irinotecan) have well-known radiosensitizing properties. Among them, we chose to examine the DNA topoisomerase I inhibitor irinotecan because: (i) its radiosensitization has been demonstrated in preclinical in vitro and in vivo studies considering high radiation dose (about 10 Gy per fraction), thus similar with SBRT regimens [19,20,21,22,23]; There are several hypotheses regarding the mechanism of interaction between Irinotecan and radiation, e.g., considering DNA damage, repair inhibition and cell cycle redistribution [24], but other mechanisms based on microvascular damage could also be implied at such high radiation dose regimen [25]; (ii) it is well tolerated in association with neoadjuvant radiation treatment and with proven efficacy for tumor downstaging [26,27,28]; (iii) it avoids the risk of cumulative Oxaliplatin-induced neuropathy; (iv) it is a central chemotherapy drug for the treatment of metastatic CRC, and has no dose-dependent toxicity [29,30].

The present study is a multicenter non-randomised phase 2 trial considering liver and/or lung oligometastatic CRC for inoperable patients who had already received at least one line of chemotherapy. Oligometastases were treated by SBRT in combination with Irinotecan in order to increase local efficacy. An ancillary study with published preliminary results [31] was also conducted, and showed that plasma ceramide—a proapoptotic sphingolipid, was rapidly generated after SBRT irradiation [25] and could be an early response predictor of SBRT. We present here the final results of the clinical efficacy and safety of this phase 2 trial.

## 2. Materials and Methods

### 2.1. Study Design and Patient Population

RADIOSTEREO-CAMPTO was a multicenter non-randomised phase 2 trial conducted in four French radiation oncology departments and registered to Clinical Trials with number NCT 01220063. Patients with liver and/or lung inoperable metastases of CRC were considered by a multidisciplinary team meeting including liver or thoracic surgeons, age ≥18 years, Eastern Cooperative Oncology Group (ECOG) performance status ≤2 and life expectancy ≥6 months, were enrolled. Metastases had to be measurable by computed tomography (CT) with the largest axis ≤60 mm (or sum of the largest axes if multiple metastases present). Hepatic magnetic resonance imaging (MRI) and/or positron emission tomography (PET) were recommended at baseline but were not mandatory. The maximum number of metastases had to be ≤3 in total (lung, liver or combined lung and liver). Patients had to have received at least one course of 5-fluorouracil (5FU)-based chemotherapy for treatment of metastases and could have received various prior lines of chemotherapy (with or without Irinotecan). Main exclusion criteria was metastases other than lung or liver even with complete radiological response.

### 2.2. Treatment Modalities and Follow-Up

At the beginning of the trial in 2008, a SBRT dose prescription (isodose surface 70–90%) of 40 Gy in 4 fractions of 10 Gy overlying the planning target volume (PTV) was mandatory, delivered twice a week over two weeks (Day 1-3-8-10). 99% of the PTV covered by less than 90% surfacic prescription (D99 < 36 Gy) was considered as a major dose deviation. A series of liver metastases treated with higher prescription doses (46–52 Gy in three fractions and up to 60 Gy in three fractions) were published in the meantime, showing a 1-year local control around 90% without notable toxicity. A dose prescription of 48 Gy in four fractions at the periphery of the PTV (D99 < 43.2 Gy as a major dose deviation), which is currently a conventional prescription dose reported in the literature [7,13], was then delivered during the second part of the trial after an amendment in 2012. According to the linear-quadratic framework (assuming α/β = 10), an effective biological dose (BED) can be computed for dose comparisons. The BED increased from 80 Gy (for 40 Gy/4 fractions) to 105.6 Gy (for 48 Gy/4 fractions) at the periphery of the PTV [7,10,11]. Additional details regarding the SBRT modalities are given in the Appendix A. 

Irinotecan (40 mg/m^2^) (Campto^®^-Pfizer, Paris, France) was intravenously injected 30–90 min before delivery of the first and third SBRT fractions (Day 1–8). The irinotecan dose used was in agreement with the reported combined neoadjuvant regimen for rectal cancers as established from previous studies [26,27,28].

Patients were followed-up clinically and with thoracoabdominal CT at 6–8 weeks, 3 months, 6 months, 9 months, 12 months and then every 6 months after SBRT. In case of a regional or distant metastatic progression outside the irradiated lesions, patients were treated according to the guidelines but were still followed up through the trial. A central reviewing of all CT images was not requested during the trial but was performed before the statistical analysis.

### 2.3. Outcomes

The primary outcome was the best local objective response rate: complete response (CR) + partial response (PR) as per the Response Evaluation Criteria in Solid Tumors (RECIST) 1.0 guidelines and determined using the CT scan of the irradiated targets. Progression of disease (PD) was also defined as per RECIST 1.0. As it can be difficult to distinguish recurrence from fibrosis/pseudoprogression after SBRT, in such cases if the RECIST 1.0 criteria for PD were met, the situation was counted as progression unless there was stability of disease (SD) as determined by follow-up images for at least 6 months. In specific cases of new or re-growth of existing lesions within or at the margin of the PTV, the lesion was considered as locally progressive. When findings were equivocal, the patient still had to be followed. If progression was confirmed at the next assessment, the date of progression assigned was the earlier date when progression was first suspected. Distant progression was defined as new lesions outside the PTV. Secondary outcomes were: early (<three months after SBRT—evaluated by the CTC-NCI 3.0 scale) and late (>three months after SBRT—evaluated by the RTOG scale) toxicity, quality of life (EORTC QLQ-C30), local control and overall survival.

### 2.4. Statistical Analysis

The statistical analysis was based on a two-stage Simon’s optimal design. Based on the literature, an objective local response rate of 75% was estimated for the SBRT irradiated targets (null hypothesis H0: P0 = 75%). Using SBRT and concomitant Irinotecan, the response rate was expected to be larger or equal to 90% (alternative hypothesis H1: Pa = 90%). An alpha risk of 10% was assumed with a statistical power of 90%. In the first stage Simon’s analysis, 16 lesions were required during stage 1: if 12 or less responses were observed, H1 would be rejected and the trial would stop. In the case of 13 or more responses, 32 additional lesions would be treated through the second stage. For the total 48 lesions, H0 would not be rejected in the case of 39 (81.25%) or less responses, and no further investigation of this treatment would be warranted, whereas H0 would be rejected in the case of 40 responses or more (P significantly > 75%). An initial recruiting time of three years was set but was then extended to three more years due to a slow recruitment rate. All the included patients were considered in the intention-to-treat (ITT) analysis.

Qualitative factors were described by means of frequency of their respective modalities and compared using of Pearson’s Chi-square test (or Fisher test). For continuous factors, independent groups were described by means of their median [range] and compared using a Student’s *t*-test (or Mann-Whitney). Baseline and post-treatment quality of life scores (at 6-, 9-month, 1-year and then every 6 months until the end of trial) were compared using a Wilcoxon signed test for matched pairs.

Survival (local progression-free [LPFS], distant progression-free [DPFS], global progression-free survival [PFS] and overall survival [OS]) were defined as follows: the time between inclusion and local or distant progression or death respectively (or date of last visit without considered event). Survival (LPFS, DPFS, PFS, OS) was described by means of Kaplan-Meier curves and compared using log-rank tests (or univariate Cox) for post-hoc analyses. CR was considered as a time-dependent variable. Median follow-up was calculated by means of inverse Kaplan-Meier method.

## 3. Results

Forty-four patients harbouring 51 metastases were recruited in four French oncology centers between 2008 and 2013 (Figure 1): thirty-four patients with liver and eleven with lung criteria mets, respectively. One patient was treated for both liver and lung metastase. One patient with two SBRT-irradiated lung lesions was lost during follow-up. The maximum bias hypothesis for this patient was used to calculate the local response within ITT: either as a local treatment failure or as a favorable local response. According to national registries, he died three years after the inclusion (patient and treatment characteristics are given in the Appendix A). Forty-three patients harbouring 49 metastases (39 liver and 10 lung targets) were analysed per the protocol (Figure 1).

### 3.1. Patient and Treatment Characteristics

Most of the patients had ECOG performance status 0 (77.3%) at inclusion (Table 1). They had a long history of CRC with a median time between diagnosis and inclusion of 29.4 months; twenty-three (52.3%) patients had synchronous metastases. Twenty-seven (61.3%) patients had received 2 or more prior lines of chemotherapy including eight (18.2%) with 3 or more.

Regarding the dose delivered, for the liver subgroup (39 targets), 29 (74.3%) and 10 (25.6%) lesions were respectively treated with the 4 × 10 Gy and 4 × 12 Gy protocols. For the lung subgroup (12 targets): eight (66.7%) and four (33.3%) lesions were treated with the 4 × 10 Gy and 4 × 12 Gy protocol, respectively. There were two and one major dose deviations in the 4 × 10 and 4 × 12 Gy protocols, respectively. Only the liver subgroup was affected.

### 3.2. Local Response and Progression-Free Survival

The objective local response rate of ITT was 86.3% (44/51—Wald 95% CI: [76.8–95.7]) or 82.4% (42/51—Wald 95% CI: [71.9–92.8]) within the maximum bias hypothesis. In per protocol analysis (49 lesions—43 patients), the objective local response was 85.7% (42/49—Wald 95% CI: [75.9–95.5]) with 26/49 (53.1%) CR, 16/49 (32.7%) PR and 7/49 (14.3%) SD; no case of progression was initially noted (Table 2). With a median follow-up of 48.9 months (Table 2 and Figure 2), the 9-month, 1- and 2-year LPFS were respectively: 84.6%, 72.5% and 67.5%; the 9-month, 1- and 2-year DPFS were respectively: 47.3%, 38.4% and 21.3%; the 1- and 2-year OS were respectively 97.3% and 72.7%.

Complete local response was significantly associated with better OS (HR 0.32, *p* = 0.02) in univariate analysis. In the liver subgroup (Figure 2), the association was not statistically significant (*p* = 0.116). The PTV volume and largest axis per metastasis > 30 mm, were associated with worst OS (HR 1.008, *p* = 0.051 and HR 2.26, *p* = 0.095, respectively) and worst local control (HR 1.012, *p* = 0.008) for PTV volume only.

For the 17 patients from the liver subgroup harbouring local CR or PR at the last evaluation, distant progression appeared for 13 (76.5%) after a median time of 6.5 months from inclusion. For the eight patients from the lung subgroup harbouring local CR or PR at the last evaluation, 3 (37.5%) had distant progression: two at 3 months and one at 9 months from inclusion.

Lung metastases were not significantly associated with better LPFS (HR 0.26, *p* = 0.199) or PFS (HR 0.51, *p* = 0.177), but with better OS (HR 0.10, *p* = 0.01) (Figure 2 and Figure 3). Patients in the lung subgroup had smaller lesions compared to liver subgroup (median PTV volume = 15.2 cm^3^ versus 45.9 cm^3^, *p* = 0.002 and median largest axis = 13 mm versus 30 mm, *p* = 0.002). A higher proportion of synchronous metastatic patients was also seen in the liver subgroup: 21/34 (61.7%) versus 2/10 (20%) (*p* = 0.031).

Neither dose protocol (4 × 10 Gy versus 4 × 12 Gy), nor synchronous versus metachronous metastases, nor number of lines of prior chemotherapy were associated with rates of local or distant progression or OS by univariate analysis.

### 3.3. Toxicity and Quality of Life

Based on CTC-NCI v3.0 evaluation, 9/43 (21%) patients (5/34 and 4/10 in the liver and lung subgroups, respectively) presented grade 2 acute reactions (see Table 3). There were no acute reactions greater than grade 2.

Two patients suffered from RTOG grade ≥ 2 late reactions (Table 4): one patient presented a grade 2 radiation pneumonitis 3 months after lung SBRT which disappeared at 6 months; one patient from the liver subgroup developed a grade 3 liver toxicity (liver enzymes) at 9 months, but in the context of a liver progression.

There was no significant worsening of the quality of life regarding the QLQ-C30 patient-based scores throughout the follow-up after treatment (9-month evaluation in Figure 4 and Appendix A).

## 4. Discussion

In this phase 2 trial, we found that high local response and low toxicity were achieved by SBRT with concomitant weekly Irinotecan administration for liver and lung inoperable CRC metastases, although the statistical limit of 90% local response with the adjunction of Irinotecan was not reached.

Firstly, local responses around 85% with 2-year local control around 70% were in agreement with the results published in the recent literature for metastatic CRC and SBRT, but most of the lesions in our study (71%) were treated with a dose regimen of 40 Gy/4 fractions overlying the target (biological effective dose BED = 80 Gy)—whereas at least BED ≥ 100 Gy is commonly associated with such outcomes in the literature [10,11,12,13,32,33]. Secondly, a majority of the patients (60.4%) had received at least two lines of prior chemotherapy—a potential factor of reduced radiosensitivity for CRC metastases [34,35,36,37]. Lastly, local evaluation after SBRT is complex and requires repeated radiological assessments over months to descriminate pseudoprogression/fibrosis from local failure [11]. Although the benefit of Irinotecan was slightly lower than the one expected in the statistical trial design, irinotecan-driven radiosensitization may have complemented our relatively lower SBRT dose prescription compared to the literature. More importantly, the combination was well-tolerated. This effect would support its use in cases where targets to be treated are close to organs at risk and the dose has to be reduced for safety reasons but dedicated studies are required to confirm such hypothesis.

Importantly, our study confirms that SBRT specifically for a metastatic CRC population has (i) a major impact on local control for liver and lung metastases, similarly with other histologies [7,8,36], while being safe and well-tolerated; (ii) comparable local efficacy to that of surgery or RFA [2,6,9]; (iii) no clear impact on distant disease progression, similarly with other local treatments [2,9]; but (iv) allows for a chemotherapy-free period of months whereas it is a short and well-tolerated treatment. A minority of patients in our study (23%) were treated for lung metastases but appeared as a single population compared to the liver patients. Significantly better survival rates confirm the better prognosis of lung metastases than non-lung [7,10,35,36], but higher local complete response rates confirm the very efficient local control particularly relevant for this population.

Because local response seems to be associated with better global outcomes [5,6,8], having reliable prediction tools for local response is relevant for the selection of patients eligible for metastases-directed treatments and in the perspective of chemotherapy-free periods. Local radiologic evaluation after SBRT classicaly required months after treatment, and may appear unsatisfactory in this regard [11]. Sophisticated imaging analysis based on machine-learning algorithms [38] could help accelerate the discrimination between fibrosis/pseudoprogression and recurrence in the future but is still far from being used in clinical practice [39]. Plasma ceramide variation with SBRT [25] is an early biomarker of response to SBRT as we have already shown in our ancillary study [31]. Such a response prediction to SBRT could help early identification of non-responder patients for which immediate post-SBRT or escalated systemic treatment would be necessary, versus responders, for which a chemotherapy-free period could be an option.

Finally, although the local response was positively associated with PFS and OS, the distant-progression risk remained high (median DPFS around 9 months), even for patients with complete local responses, and especially for patients with liver metastases, as already known from cases with exclusive surgical resection of metastases [9]. Given the low toxicity of SBRT combined with Irinotecan, a regimen consisting of peri-SBRT chemotherapy (e.g., with 5FU-Irinotecan) could be beneficial for both local and distant-disease control, and similarly with peri-operative or -RFA chemotherapy [6,9,40]. Other strong advantage of SBRT is that it could be interestingly combined with other treatments such as immunotherapy, as shown by the promising preliminary outcomes in the treatment of microsatellite instability-positive metastatic CRC [41].

Our study has several limitations. Firstly, the patients included in this study were CRC patients with either lung or liver oligometastasis, and it is known that these patients have different response profiles to treatments as well as different prognoses [1]. Secondly, the use of systemic treatments, in case of progression, could have biased the local response to SBRT. Thirdly, the impact of SBRT on OS has to be re-evaluated in situations where drugs such as anti-EGFR therapies for wild-type RAS tumors could have been used to treat progression. At the time of data collection, neither RAS nor BRAF status nor MSI testing were routinely performed and were therefore not collected in our study. However, they are important cofounding factors for both local control, PFS and OS [1,3]. Finally, patient liver MRI scans were not mandatory at baseline or during follow-up, and could have helped better evaluate the metastatic liver burden [11]. 

## 5. Conclusions

The treatment regimen of combined irinotecan and SBRT over 2 weeks for inoperable CRC metastatic patients was: (i) well tolerated; (ii) gave promising results, despite the fact that most of our patients had received multiple prior lines of treatment; (iii) could be beneficially employed for SBRT targets close to healthy sensitive tissues. Early local response prediction to SBRT would be helpful for the identification of good responders as the best candidates for chemotherapy-free periods after SBRT. 

## Figures and Tables

**Figure 1 cancers-13-00248-f001:**
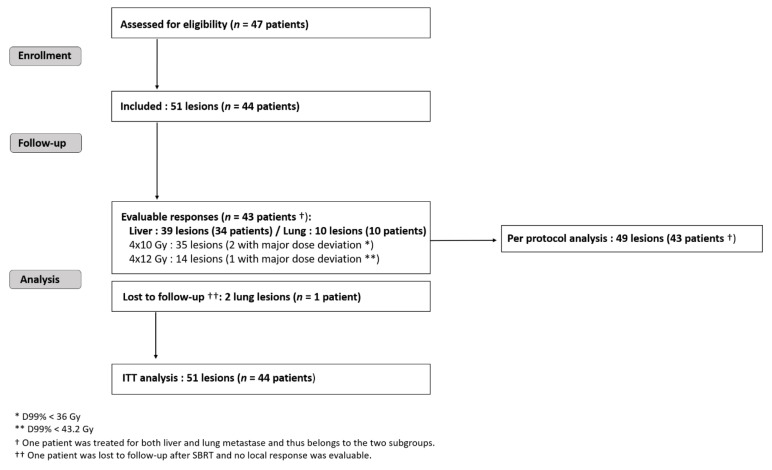
Trial flow-chart. SBRT = stereotactic body radiotherapy; RT = radiotherapy; CT = chemotherapy; ITT = intention to treat.

**Figure 2 cancers-13-00248-f002:**
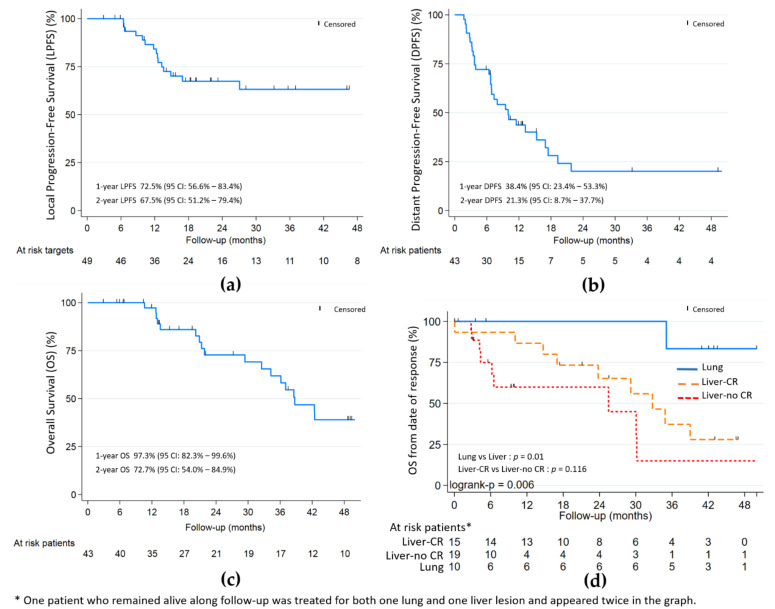
(**a**) Local progression-free (LPFS); (**b**) distant progression-free (DPFS); and (**c**) overall survival (OS). (**d**) OS by lung and liver with and without complete response (CR). Median follow-up of 48.9 months.

**Figure 3 cancers-13-00248-f003:**
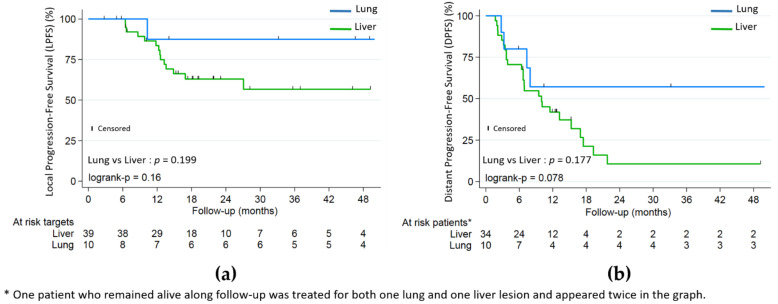
(**a**) Local progression-free (LPFS); (**b**) distant progression-free (DPFS) by lung and liver subgroups.

**Figure 4 cancers-13-00248-f004:**
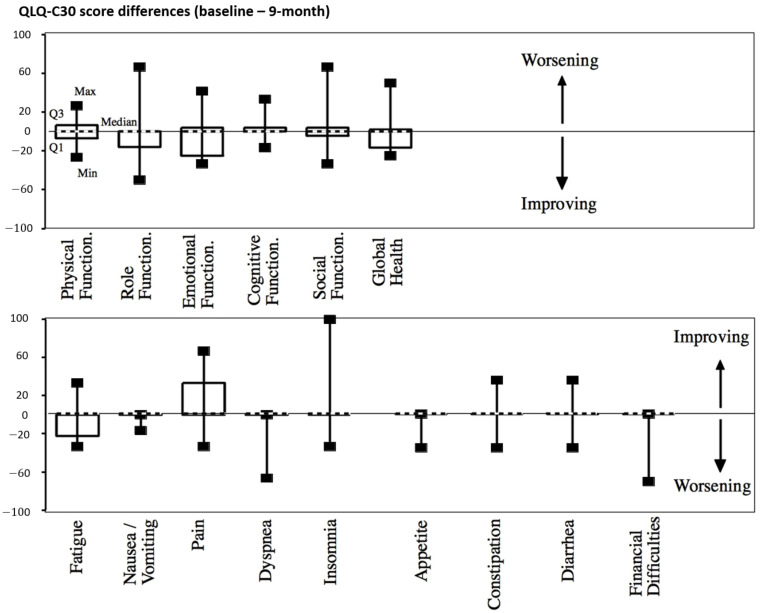
QLQ-C30 score differences between baseline and 9 months after treatment. Median values are shown with dashed lines.

**Table 1 cancers-13-00248-t001:** Baseline characteristics of patients. Median values [ranges] are shown. Synchronous = time delay with primitive diagnosis ≤4 months.

Baseline Characteristics	Liver	Lung
**Patient characteristics † (44 patients)**		
Number of patients	34 (77.3%)	11 (25%)
Median age (years)	69.4 (46–84)	67 (61–79)
Performance Status		
0	27 (61.3%)	7 (16%)
1	7 (16%)	2 (4.5%)
2	-	1 (2.3%)
Synchronous metastases with primitive diagnosis	21 (47.7%)	2 (4.5%)
Prior chemotherapy		
Adjuvant	25 (56.8%)	7 (16%)
1 metastatic line	13 (29.5%)	5 (11.3%)
2 metastatic lines	14 (31.8%)	5 (11.3%)
3 metastatic lines	5 (11.3%)	-
≥4 metastatic lines	2 (4.5%)	1 (2.3%)
prior Irinotecan	15 (34.1%)	4 (9%)
Prior metastase-directed surgery		
Liver	23 (52.3%)	1 (2.3%)
Lung	-	3 (6.8%)
Time between primitive diagnosis and inclusion (months)	28,4 (3.8–96)	32 (18–144)
Largest diameter (mm)	29 (14–57)	13 (4–44)
**Treatment characteristics**		
Number of lesions	39	12
PTV volume (cm^3^)	45,2 (7.6–241)	17 (7.7–113.2)
Surfacic isodose 4 × 10 Gy	29 (74.3%)	8 (66.7%)
Surfacic isodose 4 × 12 Gy	10 (25.6%)	4 (33.3%)
Fiducials	28 (71.8%)	4 (33.3%)
Respiratory monitoring	32 (82%)	9 (75%)

† 1 patient was treated for both lung and liver lesion.

**Table 2 cancers-13-00248-t002:** Observed response evaluation to treatment and Kaplan-Meier estimates of survival [Confidence Interval 95%].

(49 Lesions)	Complete	Partial	Stable	Progressive
Local response	26 (53%)	16 (32.6%)	7 (14.3%)	-
Last observed local response	24 (49%)	5 (10.2%)	5 (10.2%)	15 (28.5%)
Survival rates	**9-month**	**1-year**	**2-year**	**3-year**
Local Progression Free Survival	84.6% [70.3–92.3]	72.5% [56.6–83.4]	67.5% [51.2–79.4]	63% [45.3–76.4]
Distant Progression Free Survival	47.3% [31.5–61.5]	38.4% [23.4–53.3]	21.3% [8.7–37.7]	21.3% [8.7–37.7]
Overall survival	100%	97.3% [82.3–99.6]	72.7% [54–84.9]	61.8% [42.3–76.4]

**Table 3 cancers-13-00248-t003:** CTC-NCI v3.0 acute toxicity (≤3 months after treatment). A patient may present several symptoms (*N* = 43 patients).

CTC-NCI Acute Toxicity	Grade 1	Grade 2	Grade 3	Grade 4
Fatigue	5	2	-	-
Fever	-	-	-	-
Weight modifications	2	-	-	-
Anorexia	-	-	-	-
Nausea	1	3	-	-
Vomiting	1	1	-	-
Colitis	-	-	-	-
Ascites	-	-	-	-
Diarrhea	1	1	-	-
Oesophagitis	-	-	-	-
Gastritis	-	-	-	-
Peptic ulcer	-	-	-	-
Cough	2	-	-	-
Pulmonary infiltration	-	-	-	-
Abdominal pain	-	1	-	-
Liver pain	4	-	-	-
Thoracic pain	2	-	-	-
Bone pain	-	-	-	-

**Table 4 cancers-13-00248-t004:** RTOG late toxicity (>3 months after treatment). A patient may present several symptoms.

RTOG Late Toxicity	Grade 1	Grade 2	Grade 3	Grade 4
Lung	2	1	-	-
Oesophagus	1	-	-	-
Bowel	1	-	-	-
Liver	4	1	1	-
Kidney	-	-	-	-
Bone	2	-	-	-
Joint	1	-	-	-

## Data Availability

The data presented in this study are available on request from the corresponding author. The data are not publicly available due to restrictions privacy.

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
