# Peer review of "Radiosensitizing Chemotherapy (Irinotecan) with Stereotactic Body Radiation Therapy for the Treatment of Inoperable Liver and/or Lung Metastases of Colorectal Cancer"

_cancers, 2021, doi:10.3390/cancers13020248_

Round 1

Reviewer 1 Report

The article of Loïg Vaugier and colleagues entitled “Radiosensitizing Chemotherapy (Irinotecan) with Stereotactic Body Radiation Therapy for the Treatment of Inoperable Liver and/or Lung Metastases of Colorectal Cancer” is an interesting work. This work describes a multi-centre non-randomised phase 2 trial considering liver and/or lung oligometastatic CRC for inoperable patients who had already received at least one line of chemotherapy. This manuscript is well suited for publication in the journal “cancers” after minor revision.

My major concerns are:

- In the “Material and methods” part the country and state where Pfizer is located is missing.

- Figure S1 should be combined with Figure 2 and presented in the main part of the manuscript.

- Table S2 and S3 have to included into the main part of the manuscript too.

Author Response

We would like to thank the reviewer for the time and consideration in reviewing our manuscript.

  • In the “Material and Methods” part, we have corrected “Pfizer, New York” by “Pfizer, France” (line 137). We also have corrected “Pfizer, France” in the “Funding” part (line 396). We apologize for the mistake in the original manuscript.
  • We have moved Figure S1, Table S2 and S3 from the supplementary materials to the main part of the manuscript (now called Figure 3, Table 3 and 4, respectively).

With best regards,

Loig Vaugier and Emmanuel Rio

Reviewer 2 Report

This is a well-conducted study investigating the association of Stereotactic body radiotherapy (SBRT) with Irinotecan in the treatment of liver and lung metastases from colorectal cancer. The manuscript is well written and stands as an original contribution to the poorly explored field of SBRT and chemotherapy integration.

BROAD COMMENTS

1 - Although in this context irinotecan is considered as a radiosensitizing agent, it should be considered the hypothesis that observed differences might be a consequence of a simply additive effect. The references about chemotherapy as a radiosensitizing agent in radiation treatment of liver metastases (16,17,18) all deal with non-SBRT regimens with small fraction doses. As in high fraction doses of SBRT, different radiobiological mechanisms (such as radiation-induced vascular damage) may play a major role,  the authors should provide some hypothesis to explain the interaction of Irinotecan and SBRT in their regimen.

2 – In their argument about benefits of Irinotecan-SBRT combination for targets close to organs at risk, the authors should be careful not to overestimate the strenght of evidence from their data. That benefit should require a differential effect between tumor and healthy tissues which seems to be unproven at the present time.

SPECIFIC COMMENTS

1 - Response rate as a primary endpoint : The timing of evaluation of response rate is not clearly defined. The authors should specify whether it is at a fixed time point after treatment or the best response achieved at any time.

Author Response

We would thank the reviewer for the time and consideration in reviewing our manuscript.

  • We fully agree with the point that observed results of the combination of Irinotecan and SBRT might be a consequence of an additive effect. Published clinical studies have explored conventional radiation doses with chemotherapy as highlighted by the reviewer and references (16, 17, 18) but in vitro and in vivo studies also have corroborated the radiosensitizing properties of Irinotecan with high radiation dose per fraction (commonly 10 Gy), thus similar with SBRT regimens. It is true that the exact mechanisms of interaction are unknown. Hypotheses based on DNA damage, repair inhibition or cell cycle redistribution, have been investigated but other mechanisms based on vascular radiosensitivity (more specific to such high radiation doses) could be part of it. One objective of our ancillary study (doi:10.1016/j.radonc.2016.03.014) considering the dosing of ceramides (proapoptotic sphingolipid) as a potential early biomarker of response, has consisted in fact in exploring such assumption. We have added/modified the following sentences in the Introduction (line 81-87) in that sense: “its radiosensitization has been demonstrated in preclinical in vitro and in vivo studies considering high radiation dose (about 10 Gy per fraction), thus similar with SBRT regimens [19] [20] [21] [22] [23]; There are several hypotheses regarding the mechanism of interaction between Irinotecan and radiation, e.g. considering DNA damage, repair inhibition and cell cycle redistribution [24], but other mechanisms based on microvascular damage could also be implied at such high radiation dose regimen [25] ». References 22 (doi:10.1016/s0167-8140(97)01924-5), 23 (doi:10.1111/j.1349-7006.1997.tb00369.x), 24 (PMID: 11497228) and 25 (doi:10.1126/science.1082504) have been added.
  • We fully agree with the fact that caution is needed before concluding to the benefits of Irinotecan-SBRT combination for targets close to organs at risk and that our sentence has to be qualified. We have modified the text in that sense by “Irinotecan-driven radiosensitization may have complemented” (line 334), “This effect would support its use » and we have added « but dedicated studies are required to confirm such hypothesis.” (line 335-338)
  • We apologize for the uncertainty in the definition. We have added “The primary outcome was the best local objective response rate » (line 151).

With best regards,

Loig Vaugier and Emmanuel Rio